🔓 | **Open Peer Review** | Environmental Microbiology | Research Article

# Using genomics to explore the epidemiology of vancomycin resistance in a sewage system

Emilie Egholm Bruun Jensen,[1] Saria Otani,[1] Ivan Liachko,[2] Benjamin Auch,[2] Frank M. Aarestrup[1]

**ABSTRACT** *VanHAX*-mediated glycopeptide resistance has been consistently high in one of the three main sewer systems in Copenhagen, Lynetten, for +20 years. To explore this for other glycopeptide resistance genes, and whether the colonization has resulted in establishment of multiple bacterial taxa, we mapped 505 shotgun metagenomic data sets from the inlet of three sewage treatment plants to 831 different glycopeptide resistance genes. Only *vanHAX* and *vanHBX* genes were differentially abundant in Lynetten. Analyses of eight contigs suggested limited variations in the flanking regions. Proximity ligation metagenomic analysis of 12 samples from Lynetten identified 441 and 5 paired reads mapping to *vanHAX* and *vanHBX*, respectively. The other end of these reads was mapped to generated metagenomic-assembled genomes and NCBI using BLAST. *vanHBX* could only be linked to the phylum level (Bacillota). Plasmid analysis of *vanHBX* Hi-C contigs showed that these were mainly located on plasmids reported found in enterococci species. Most *vanHAX*-linked reads could only be linked to phylum and class level, but some reads were assigned to *Enterococcus faecium* (7 reads), *Enterococcus faecalis* (4 reads), *Paenibacillus apiarius* (2 reads), and *Paenibacillus thiaminolyticus* (27 reads). Ten of the 20 Hi-C contigs-containing *vanHAX* were annotated as plasmid, all reported found in *Enterococcus* species. This study shows that while Hi-C technology is valuable for linking antimicrobial resistance genes to bacterial taxa, it suffers from challenges in reliably mapping the linked read to a genomic region with sufficient taxonomic information. Our results also suggest that over the +20 years of colonizing a sewer system, *vanHAX* has not become widespread across multiple taxa, remaining primarily in *E. faecalis* and *E. faecium*, with the exception of *Paenibacillus*.

**IMPORTANCE** Long-term colonization of microbial communities with antimicrobial-resistant bacteria is expected to result in sharing of the resistance genes between several different bacterial taxa of the communities. We investigated microbiomes from a sewer, which have been colonized with glycopeptide-resistant bacteria harboring the mobile *vanHAX* gene cluster for a minimum of 20 years, using metagenomics sequencing and Hi-C. We found that despite the long-term presence in the sewer, the *vanHAX* genes have seemingly not disseminated widely.

**KEYWORDS** Hi-C, metagenomics, metagenome assemblies, antimicrobial resistance, sewage, glycopeptide resistance

Wastewater-based surveillance of antimicrobial resistance has gained increased interest in recent years and shown great potential in both showing trends in the human populations connected to the sewer as well as understanding the microbial dynamics in the sewer (1–8). Different approaches have been explored including phenotypic testing of culturable bacteria or quantification of antimicrobial resistance genes (ARGs) using different genomic approaches such as qPCR, micro-arrays and metagenomics (8–11). However, a common limitation of the genomic approaches is

**Peer Reviewer** Benjamin Cole Davis, Virginia Polytechnic Institute and State University, Blacksburg, Virginia, USA

Address correspondence to Frank M. Aarestrup, fmaa@food.dtu.dk.

The authors declare no conflict of interest.

See the funding table on p. 12.

that it has been difficult to determine bacterial host(s) of the ARGs that are carried on extrachromosomal elements (12). Thus, in recent years, different approaches have been explored including Hi-C and single-cell sequencing (13–15).

Conventional metagenomics captures the entire DNA content of a sample indiscriminately, making it difficult to trace the bacterial origin of specific DNA sequences (16, 17). In contrast, Hi-C utilizes a proximity-ligation approach, which relies on the co-localization of DNA within a single cell. By cross-linking molecules in close proximity (within the same bacterial cell), Hi-C can infer the bacterial origin of plasmids or ARGs (13–15, 18, 19).

During a longitudinal surveillance study of three different inlet sewage treatment plants in Copenhagen, Denmark, we observed a consistent and very frequent occurrence of glycopeptide resistance genes in one of the systems (Lynetten) (2). A producer of antibiotics was located in the catchment area, only a few kilometers upstream from the inlet and according to their website, they produced vancomycin hydrochloride at their site in Copenhagen (https://xellia.com/products/active-pharmaceutical-ingredients). Previous studies using enterococci enriching media found a frequent presence of vancomycin-resistant enterococci, as well as dissemination of the *vanA* gene among different enterococcal species (20, 21). However, the transmission, host range, or epidemiology of other vancomycin resistance genes were not investigated. In Denmark, clinical surveillance of vancomycin-resistant enterococci has shown the highest prevalence in the Capital region, predominantly being *Enterococcus faecium* and revealed a shift from *vanHAX* to *vanHBX* clusters from 2015 to 2022 (22).

Resistance to glycopeptide is typically categorized into *vanA*-type and *vanC*-type resistance (23). The genes encoding *vanA*-type glycopeptide resistance are organized in a cluster of three slightly overlapping genes (*HAX*) encoding reactions that all are important for the phenotypic resistance, with two upstream regulatory genes and two downstream accessory genes. The entire gene cluster is often found located on transposons. The genes encoding *vanC*-type resistance are also organized with three genes (*C-XY-T*) and two downstream regulatory genes. Different glycopeptide resistance genes have been found widespread in different environmental bacteria including the glycopeptide-producing organisms (24–28). To date, a very large number of glycopeptide resistance genes have been detected (29), but only a subset of these are considered to have been mobilized from their chromosomal locations (30).

Because of the observed comparatively high abundance of *vanHAX* in Lynetten, we investigated whether this was also the case for other glycopeptide resistance genes only identified through functional cloning. Furthermore, we hypothesized that during the long-term colonization of the Lynetten sewer substantial evolution and horizontal gene transfer would have taken place. To investigate this, we combined both conventional metagenomics and Hi-C technologies to perform an in-depth analysis of different glycopeptide resistance gene abundances over time in waste-water samples collected over 7 years in Copenhagen. We also analyzed the gene synteny of the most common glycopeptide resistance genes and the taxonomical link as determined by Hi-C.

## MATERIALS AND METHODS

### Samples and data

Samples collection and sequencing of 324 samples from Rensningsanlæg Avedøre (56 samples), Rensningsanlæg Damhusåen (148 samples), and Rensningsanlæg Lynetten (120 samples) from November 2015 to November 2018 have previously been published (2) and are available at the European Nucleotide Archive (ENA) under accession number PRJEB34633 and PRJEB13832. An additional 181 samples from Rensningsanlæg Avedøre (61), Rensningsanlæg Damhusåen (59), and Rensningsanlæg Lynetten (61) were collected and processed as previously described (31). Briefly, 1–2 L of unprocessed, non-filtered, and untreated urban sewage was collected over 24 h. About 500 mL from each sample was centrifuged for 10 min at 10,000 × *g* to collect the sewage pellets, followed by DNA extraction as previously described (2). Sequencing data are available

under ENA bioproject PRJEB68319 and PRJEB79372. Table S1 contains the ENA run accession for each sample.

## Hi-C preparation and sequencing

For this project, additional data were generated using Hi-C for 12 samples from RL. From each sample, a total of 250 mg of sewage pelleted sludge was prepared following the ProxiMeta Hi-C Kit Protocol version 4.0, supplied by Phase Genomics (Seattle, USA). Sewage sludge was suspended in a formaldehyde solution to achieve a crosslinking concentration of 1% vol. This mixture was then incubated at room temperature for 20 min, with gentle inversions every 5 min. Crosslinking was stopped by the addition of glycine (ProxiMeta Hi-C Kit), to achieve a final concentration of 1% vol. This mixture was then incubated at room temperature for an additional 20 min, with occasional stirring to homogenize the mixture. Enzymatic digestion of the cross-linked DNA in each sample was done with Sau3AI and MluCI enzymes (ProxiMeta Hi-C Kit). The digested DNA fragments were then proximity-ligated with biotinylated nucleotides. This step facilitated the formation of chimeric DNA molecules comprising sequences from different genomic regions that were in close proximity to the original cellular structure. These chimeric molecules were then purified using streptavidin beads and processed further using the library preparation reagents from ProxiMeta. All DNA was extracted with a ZYMObiomics DNA miniprep kit and prepared using ProxiMeta library preparation reagents. Sequencing was performed on an Illumina NovaSeq X generating 150 bp paired-end reads.

## Preprocessing and mapping of sequencing reads

All the Hi-C shotgun reads were filtered and trimmed for quality using fastp using default parameters (32), and then assembled with MEGAHIT (33) using default options. All the conventional metagenomic trimmed sequencing data previously described (2, 31) were re-mapped for this study. Read assignment was performed with KMA version 1.4.2 (34) (with the following flags: -mem_mode -ef -1t1 -cge) against the glycopeptide resistance genes from our recently created PanRes database (29) for assignment of ARGs. The PanRes database includes 14,078 unique ARGs and is a combination of several existing ARG databases into one. Here, we further explored glycopeptide ARGs mapping to ResFinder (30) which includes ARGs that are assumed to have been mobilized, and to ResFinderFG (35) which includes ARGs identified only by functional cloning. Both ResFinder and ResFinderFG are part of the PanRes database. The following hierarchy reduction of the PanRes database was used: ResFinder > ResFinderFG; that is, if a gene was present in both of the sub-databases, it was annotated as a ResFinder gene. If it was present in only the ResFinderFG database, it was annotated as a ResFinderFG gene.

## Compositional data analysis

The read fragment counts of the different glycopeptide resistance genes obtained from the KMA mapping against the PanRes database (29) were used as the gene counts for the downstream analysis. First, the counts were normalzed by the length of the reference divided by 1,000 to avoid small numbers interfering with the zero replacement of the compositional data analysis. Due to the compositional nature of microbiome data (36), zero replacement was performed on the normalized counts before doing the centered-log ratio (clr) transformation (Equation 1). Here, a sample is a composition $x$ of $D$ parts. Where $x_i$ is a count, for example, read resistance gene count, and $D$ represents the parts, for example, resistance genes.

$x = [x_1, \ldots, x_D]$

$$\mathrm{clr}(x_1, ..., x_D) = \left( \log\left(\frac{x_1}{G(x)}\right), ..., \log\left(\frac{x_D}{G(x)}\right) \right), \quad (1)$$

where $G(x) = {}^D\sqrt{x_1 \cdot, ..., \cdot x_D}$

## Gene synteny and flanking regions

The assembled conventional metagenomic contigs were used to study the gene synteny, that is, the flanking regions of *vanHAX_2_m97297* (pan_6936 in PanRes database) and *vanHBX_1_af192329* (pan_7222 in PanRes database) gene clusters using the Flankophile pipeline (37). Relevant thresholds for upstream and downstream flanking regions were chosen based on the length of the contigs containing the target sequence to include as much flank as possible, see Table S2 for the thresholds used.

## Linking vancomycin resistance to genomes

### Linking contigs to conventional metagenomic-assembled genomes

The trimmed reads were assembled and binned as previously described (31). Briefly, assembly was performed with MEGAHIT version 1.2.9 (33) with default parameters, and binning was performed with MetaBAT2 version 2.15 (38). Contigs containing glycopeptide resistance genes were identified by mapping with KMA version 1.4.2 (34) against the two vancomycin resistance genes of interest, *vanHAX* and *vanHBX*, from the PanRes database (29). Quality of the genome bins was evaluated using CheckM lineage_wf (39). If the identified contig belonged to either a high quality (HQ: bin ≥ 90% complete and bin ≤ 5% contamination) or a medium quality (MQ: bin ≥ 70% complete and bin ≤ 10% contamination) bin, taxonomic annotation was performed with the GTDB-Tk (gtdbtk classify_wf) using the Genome Taxonomy Database (40). These MQ and HQ metagenomic assembled genomes (MAGs) were placed in the GTDB-Tk reference tree, and visualized and annotated using the library ggtree version 3.8.2 (41) within R version 4.3.0.

### Linking Hi-C reads to Hi-C metagenomic clusters

Deconvolution of the sewage metagenomes was carried out using the ProxiMeta workflow (42), which combines Hi-C data within conventional shotgun sequencing assemblies to obtain Hi-C metagenomic clusters. The 12 Lynetten samples were further investigated using this Hi-C metagenomic sequencing. Hi-C contigs containing *vanHAX* and *vanHBX* resistance genes were identified using KMA version 1.4.2 (34) against these two glycopeptide resistance genes of interest. Quality of the Hi-C clusters was evaluated by CheckM lineage_wf (39) and decontamination of the Hi-C clusters was performed with MagPurify version 2.1.2 (43) with the following positional arguments: coverage, gc-content, known-contam, phylo-markers, and tetra-freq.

### Linking Hi-C reads to conventional MAGs

For each sample, the Hi-C reads containing one of the two vancomycin resistance genes of interest (*vanHAX* and *vanHBX*) were mapped against the conventional MAGs of high and medium quality belonging to that sample using BBMap version 36.49 (https://sourceforge.net/projects/bbmap/) (44). This was to include information from all Hi-C reads, and not limit the taxonomic analysis to only those Hi-C reads being binned in Hi-C clusters. Taxonomic annotation was performed using GTDB-Tk (40).

### Linking vancomycin resistance to NCBI taxa

Rather than restricting the analysis solely to the MAGs and Hi-C clusters, a read- and contig-based approach was applied to include additional information. The assembled contigs from conventional metagenomic sequencing were analysed separately from the Hi-C reads themselves. First, the sequences were mapped against the PanRes database (29) using KMA version 1.4.2 (34) with hierarchy reduction: ResFinder > ResFinderFG as previously described. Taxonomic annotation of the sequence (being either Hi-C read or conventional contig) containing either *vanHAX* or *vanHBX* was carried out using BLASTN version 2.13.0 (45). Only BLASTN hits with high confidence were considered (*e*-value <1e−50). The taxIDs were mapped to their taxonomy according to the NCBI

taxonomy database (taxdmp.zip downloaded on 18. September 2023 from: https://ftp.ncbi.nlm.nih.gov/pub/taxonomy/). The taxonomy of each sequence (Hi-C read or conventional contig) was determined by the lowest common tax level identified among the BLASTN hits. that is, if multiple hits pointed to different species, but they agreed on the genus level, the genus level was the lowest common tax level for this sequence. Ambiguities in the taxonomic classification of the vancomycin-containing contigs were attempted to be resolved with MMseqs2 (46) taxonomy workflow (release 15-6f452) using the lowest common ancestor approach (--lca-mode 4) with the GTDB database version 220 (47).

Known gene homologous to enterococcal *vanA* and *vanB* have been reported in some *Paenibacillus* species (27). To ensure sequences mapped to vancomycin resistance genes were *van*-genes and not *van*-related ligases, these sequences were investigated further: sequences mapping to either *P. apiarius* or *P. thiaminolyticus* were re-mapped with BLASTN against the known gene homologous with Genbank accession AY648698 (*P. apiarius*) and AY648035 (*P. thiaminolyticus*), as well as the *vanHAX* gene from the PanRes database (pan_6936).

## Plasmid annotation and characterization of Hi-C assembled plasmids, Hi-C assembled contigs, and conventional contigs, all containing vanHAX and vanHBX resistance genes

KMA version 1.4.2 (34) was used to identify *vanHAX* and *vanHBX* resistance genes (pan_6936 and pan_7222 in PanRes database) within the sequences (being either Hi-C assembled plasmids from the ProxiMeta platform (40), Hi-C assembled contigs from the ProxiMeta platform (40), or conventional contigs). Plasmid annotation and characterization was attempted with PlasmidFinder version 2.1 (48), and Platon version 1.7 (49).

## RESULTS

### Conventional metagenomic analyses

From the 181 metagenomic sequenced sewage samples from Lynetten, a total of 3.699 billion paired-end reads were obtained (average: 20.44 million paired-end reads per sample, range: 36,329–64,473,542 paired-end reads per sample, standard deviation: 18.26 million) (Table S1). On average, 0.00006% of the reads mapped to the glycopeptide resistance genes from the ResFinder and ResFinderFG subset database from the original PanRes database, and on average, 0.00173% mapped to the entire PanRes database.

### *Glycopeptide resistance profiles using the conventional metagenomics*

The occurrence of glycopeptide resistance in sewage from the Lynetten treatment plant in Copenhagen was compositionally stable over time (Fig. S1). In total, 39 different glycopeptide resistance genes from the ResFinder database and 792 from the ResFinderFG database were identified within the sewage samples from the Lynetten treatment plant in Copenhagen. Table 1 shows the top 5 most abundant glycopeptide resistance genes from the ResFinder database and Table 2 shows the top 10 most abundant ResFinderFG glycopeptide resistance genes. The most abundant of these genes are found in the ResFinderFG database.

When comparing Lynetten sewage resistome to the two remaining treatment plants in Copenhagen: 117 sewage samples from Avedøre and 207 samples from Damhusåen,

**TABLE 1**  Top five most abundant ResFinder glycopeptide resistance genes in Lynetten sewage.

|  | clr median | clr variance |
| --- | --- | --- |
| resfinder\|vanhax_2_m97297 | 2.668 | 2.291 |
| resfinder\|vanhbx_1_af192329 | 0.718 | 0.487 |
| resfinder\|vang2xy_1_fj872410 | 0.300 | 0.307 |
| resfinder\|vanhax_pa_1_dq018711 | 0.062 | 0.430 |
| resfinder\|vanhfx_1_af155139 | 0.018 | 0.485 |

**TABLE 2**  Top 10 most abundant ResFinderFG glycopeptide resistance genes in Lynetten sewage.

|  | clr median | clr variance |
|---|---|---|
| functional_amr\|van-ligase\|kf627618.1\|pediatric_fecal_sample\|cyc | 2.964 | 0.873 |
| functional_amr\|van-ligase\|kf629262.1\|pediatric_fecal_sample\|cyc | 2.747 | 1.313 |
| functional_amr\|van-ligase\|kf628295.1\|pediatric_fecal_sample\|cyc | 2.714 | 1.091 |
| functional_amr\|van-ligase\|kf629426.1\|pediatric_fecal_sample\|cyc | 2.709 | 1.345 |
| functional_amr\|van-ligase\|kf629150.1\|pediatric_fecal_sample\|cyc | 2.512 | 1.237 |
| functional_amr\|van-ligase\|kf628408.1\|pediatric_fecal_sample\|cyc | 2.489 | 1.528 |
| functional_amr\|van-ligase\|kf629844.1\|pediatric_fecal_sample\|cyc | 2.480 | 1.048 |
| functional_amr\|van-ligase\|kf629505.1\|pediatric_fecal_sample\|cyc | 2.394 | 2.080 |
| functional_amr\|van-ligase\|kf626830.1\|pediatric_fecal_sample\|cyc | 2.283 | 1.122 |
| functional_amr\|van-ligase\|kf629216.1\|pediatric_fecal_sample\|cyc | 2.273 | 0.878 |

compositional differences were observed (Fig. 1A) (PERMANOVA, *P* value = 0.001). The resistomes from Lynetten were more similar to each other compared to the other treatment plant ARGs (Fig. 1A). It was evident that the only glycopeptide resistance genes being differentially abundant were the ones belonging to the *vanHAX* and *vanHBX* clusters (Fig. 1B and C).

### Glycopeptide gene synteny

The gene synteny analysis of *vanHAX* and *vanHBX* gene clusters (Fig. 2) was performed with relevant thresholds on the upstream and downstream flanking region to include as much flank and as many sequences as possible. The *vanHAX* gene cluster was identified in 22 contigs and the same gene variant was found across all contigs (Table S2). Six *vanHAX*-containing sequences passed a flank threshold of 6,056 bp downstream and 1,875 bp upstream from the gene cluster (Fig. 2A). Here, transposon *Tn4430* was found in all flanking regions, in addition to the H inversion gene, *hin*, and the response regulator, *walR*. Phylogeny based on the flanking regions of the *vanHAX* cluster, showed two clusters: One containing *vanY* and one containing the transposase *ISBcy1* just upstream of the target gene cluster. The longest contig containing *vanHAX* had a length of 16,677 bp, with a 4,584 bp upstream flank and a 9,486 bp downstream flank (Fig. S2A).

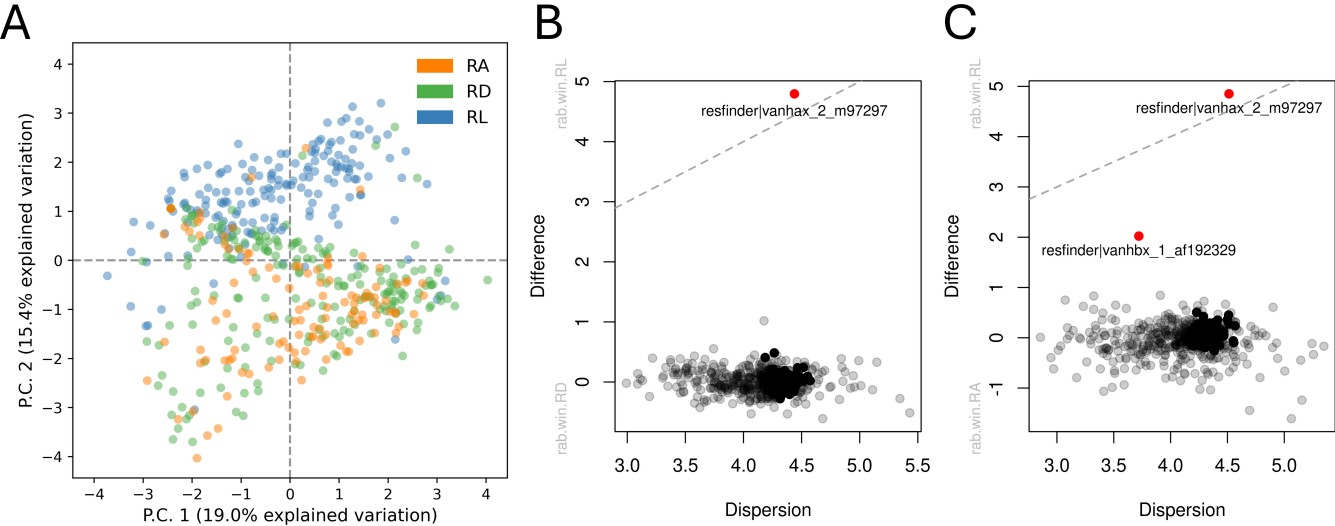

**FIG 1**  (A) Principal component analysis based on abundances of the ResFinder, ResFinderFG, and ResFinderNG glycopeptide resistance genes showing the clustering of samples from the three different sewage treatment plants: Lynetten (RL), Avedøre (RA), and Damhusåen (RD). The ordination analysis is based on a threshold on the feature clr variance and feature clr median across samples (feature clr variance >2 and feature clr median >0.5) to include only abundant and relevant features for the clustering. (B) Differential abundant glycopeptide resistance genes comparing Rensningsanlæg Avedøre (RA) and Rensningsanlæg Lynetten (RL). (C) Differential abundant glycopeptide resistance genes comparing Rensningsanlæg Damhusåen (RD) and Rensningsanlæg Lynetten (RL).

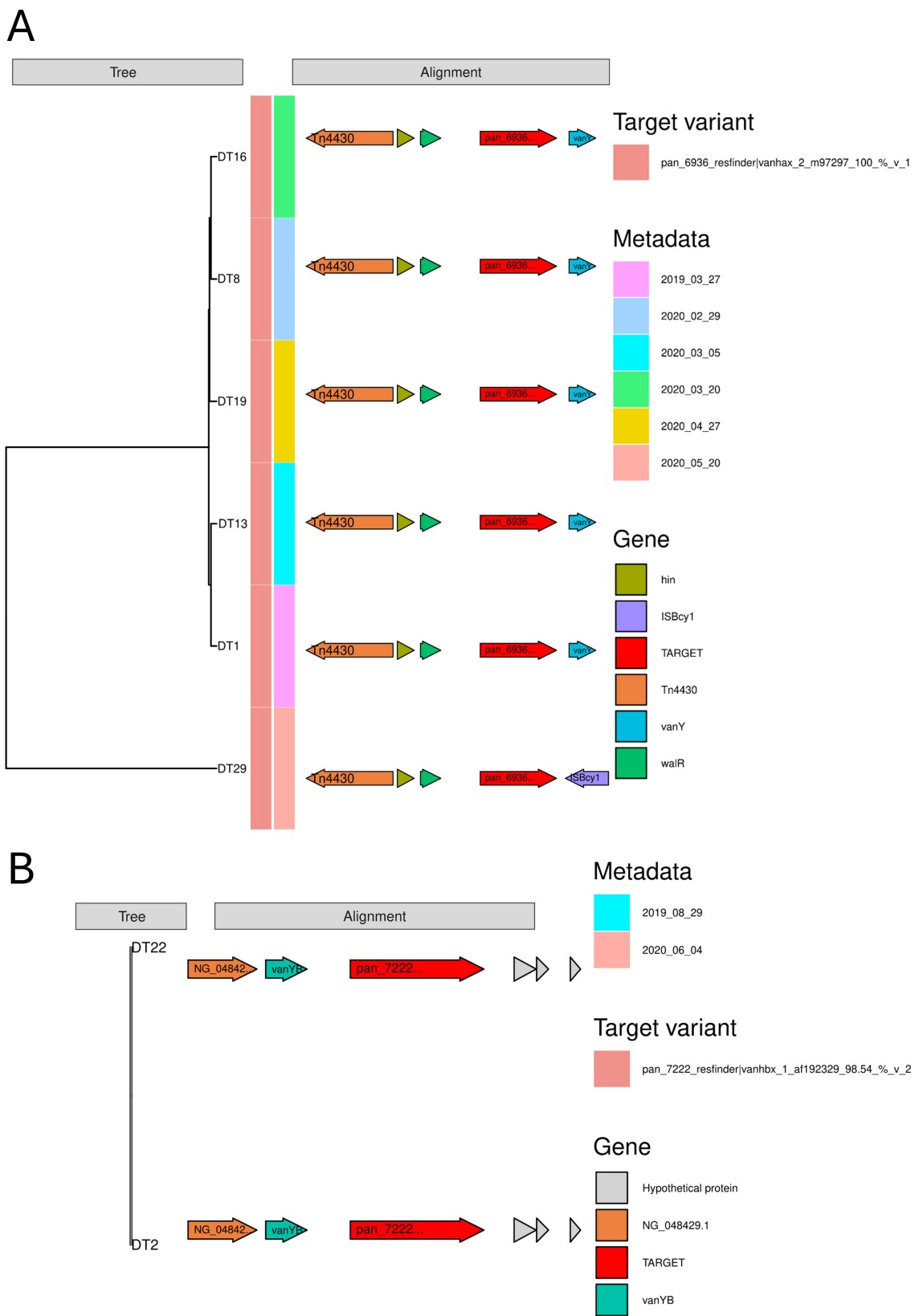

**FIG 2** (A) Flanking analysis of the *vanHAX* cluster: 6,056 bp downstream and 1,875 bp upstream from the gene. The distance tree is based on the alignment of the flanking regions. (B) Flanking analysis of the *vanHBX* cluster: 3,196 bp downstream and 2,363 bp upstream of the gene. The distance tree is based on the alignment of the flanking regions.

The *vanHBX* was identified in two contigs (Table S3) and visualized with a 1,875 bp upstream flank and a 6,056 bp downstream flank (Fig. 2B). The same target variant was identified in both, and the clustering of the flanking region showed no difference. The vancomycin resistance genes *vanSB* (NG_048429.1) and *vanYB* were identified upstream of *vanHBX*. The longest *vanHBX* containing contig was 31,543 bp with a 25,457 bp upstream flank and 3,480 bp downstream flank (Fig. S2B), and showed an *Int-Tn* transposase downstream from the target cluster.

### Linking vancomycin resistance to MAGs

From the 4,276 Hi-C bins, 18 bins were identified as having a contig containing the *vanHAX* gene cluster (Table S4). Of these, three were of high quality (HQ: bin ≥ 90% complete and bin ≤ 5% contamination) and two were of medium quality (MQ: bin ≥ 70% complete and bin ≤ 10% contamination). Two bins were identified as having a contig where the *vanHBX* gene cluster was present. One of these was of medium quality. Three of the five MAGs containing *vanHAX* were from the *Streptococcus* genus, one was from the *Trichococcus* genus and one was from the *UPXZ01* genus (*Paludibacteraceae* family) (Fig. 3). The *vanHBX* gene cluster was found in a bacteria from the *Bacteroides* genus. It cannot be excluded that they mapped to regions conserved across *Lactobacillales*.

### Linking vancomycin resistance to NCBI taxa using a read-based approach

Instead of limiting the analysis to the MAGs, the information from all assembled contigs were considered. Using the lowest common tax level approach (see Materials and Methods), the taxonomy of the identified 40 *vanHAX*- and 38 *vanHBX*-containing contigs was identified (Fig. 4; Tables S5 and S6). Most of the assigned contigs harboring the *vanHAX* gene cluster could only be assigned at the phylum level; however, two contigs were assigned as *E. faecium* and two were assigned *Enterococcus C saigonensis*. For *vanHBX*, eight of the contigs were assigned as *E. faecium*. It was not possible to compare the taxonomic assignment across methods (contig vs MAG), since none of the *vanHAX*- or *vanHBX*-containing contigs assigned at the species level belonged to a medium- or high-quality bin (Tables S7 and S8).

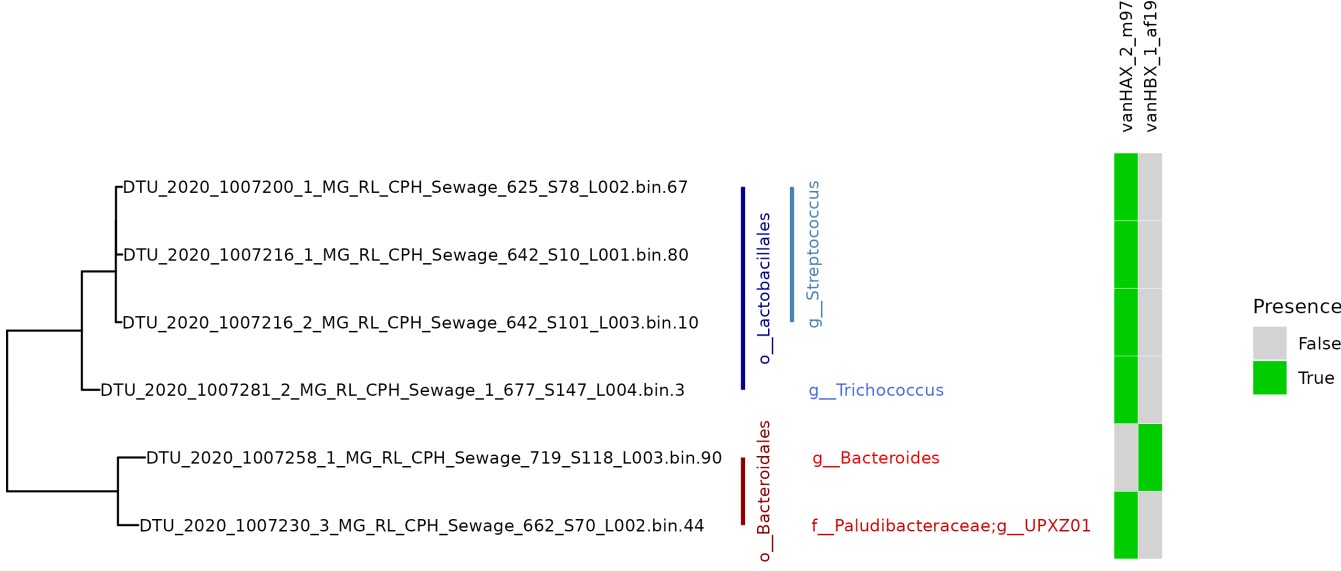

**FIG 3** Phylogeny of the high- and medium-quality MAGs containing either *vanHAX* or *vanHBX* resistance (HQ: bin ≥ 90% complete and bin ≤ 5% contamination, MQ: bin ≥ 70% complete and bin ≤10% contamination) in the GTDB_Tk reference tree.

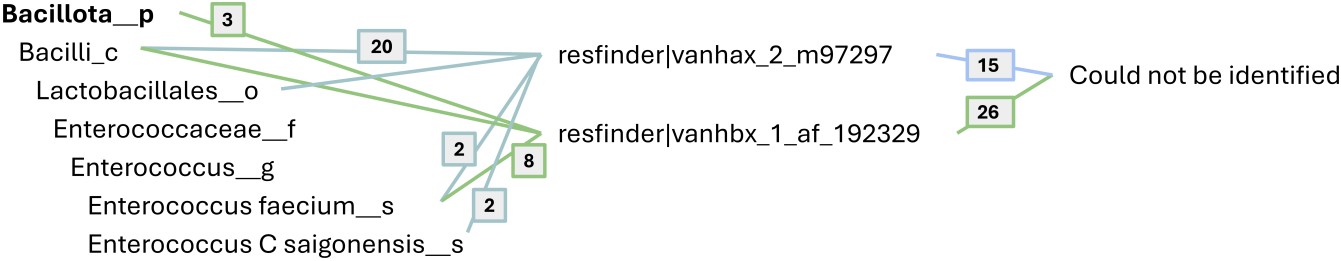

**FIG 4** Taxonomy of the 40 *vanHAX*- and 38 *vanHBX*-containing conventional contigs using a BLASTN lowest common tax level approach. Only hits with high confidence were considered for this approach (*e*-value <1e−50).

## Linking vancomycin resistance to Hi-C clusters

The ProxiMeta deconvolution platform was able to produce a total of 2,323 Hi-C clusters from the 12 samples. About 197 of these were of high quality and 669 were of medium quality. The *vanHAX* gene cluster was identified in 20 contigs and the *vanHBX* gene cluster was identified in 22 contigs (Table S10). Five of these *vanHAX*-containing contigs were associated with a cluster, while none of the *vanHBX*-containing contigs were associated with a cluster. Quality assessment of the clusters containing *vanHAX* showed that none of the identified clutters were of high or medium quality (Table S11A). Three of the four clusters were highly contaminated (range: 37–85%) and one of the remaining clusters was highly incomplete (28%). Decontamination lowered the contamination of the three highly contaminated clusters (range after decontamination: 28–43%) (Table S11B). However, after purification, the four Hi-C clusters were still not of high enough quality, and the CheckM marker lineage classification did not improve.

## Linking Hi-C reads containing vancomycin resistance to conventional MAGs

Information from all Hi-C reads were considered to avoid limiting the analysis to the identified Hi-C clusters. Within the 12 samples being Hi-C sequenced, *vanHAX* were identified in 441 of the Hi-C reads and *vanHBX* was identified in 5. These reads were mapped to the high- and medium-quality MAGs to link the resistance genes to bacterial taxa. Only reads from two samples mapped to MAGs originating from the same sample (Table S12). Two MAGs were identified, one in each sample, and both were annotated as *Streptococcus parasuis*.

## Linking Hi-C reads containing vancomycin resistance to NCBI taxa

The 441 Hi-C reads containing *vanHAX* and the 5 Hi-C reads containing *vanHBX* were linked to their bacterial taxa using a BLASTN lowest common tax level approach (see Materials and Methods). By only considering hits of high confidence (*e*-value <1e−50), we were able to link the reads to five bacterial taxa (Fig. 5). Both *E. faecium* and *E. faecalis* were identified harboring *vanHAX*, but also *P. apiarius* and *P. thiaminolyticus*.

To ensure that *vanHAX* reads mapping to *P. apiarius* and *P. thiaminolyticus* were not the *van*-related ligases known to be part of the genome of these species, these reads were mapped to both *vanHAX*- and *van*-related ligases. Using BLASTN with *e*-value <1e−20, the reads only mapped *vanHAX*.

## Plasmid annotation of assembled Hi-C plasmids, assembled Hi-C contigs, and conventional contigs, all containing vanHAX and vanHBX resistance genes

Within the assembled Hi-C plasmids, *vanHAX* was identified within 10 contigs, and *vanHBX* was identified within 7 contigs. After the assembled plasmids were annotated, *vanHAX* were associated with seven unique plasmids, mainly found in *Enterococcus* species, and *vanHBX* were associated with three unique plasmids, all reported in *E. faecium* (Table S13). Plasmid characterization returned no hits with PlasmidFinder (48).

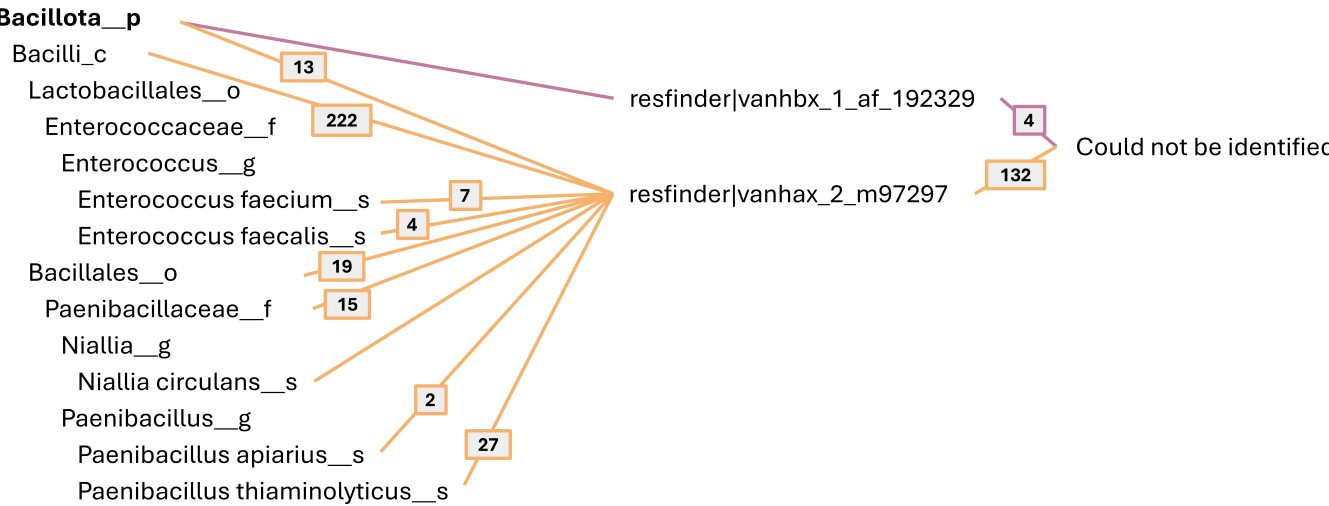

**FIG 5** Taxonomy of the 441 *vanHAX*- and 5 *vanHBX*-containing Hi-C reads using a BLASTN lowest common tax level approach. Only hits with high confidence were considered (*e*-value <1e−50).

The contigs were validated as plasmids with Platon (49); however, no known replication genes, mobilization genes, *oriT* sequences or conjugation genes were identified.

Expanding the analysis to the 20 *vanHAX*-containing Hi-C assembled contigs confirmed that 10 contigs were plasmids and 10 were not (Table S14). All 22 of the *vanHBX*-containing Hi-C assembled contigs were annotated as plasmids reported in either *E. faecalis* or *E. faecium* (Table S14).

Repeating the plasmid analysis for the 40 conventional contigs containing *vanHAX* and the 38 conventional contigs containing *vanHBX* yielded results consistent with the findings from the Hi-C analysis (Table S15).

## DISCUSSION

Here, we used conventional metagenomics and Hi-C of selected sewage samples to study the abundance, genomic diversity, and taxonomical location of glycopeptide resistance in a sewer with a high abundance of glycopeptide resistance (2, 20), that likely originated from the former glycopeptide-producing factory located upstream.

Genes encoding glycopeptide resistance have previously been found widespread in different environmental bacteria including the glycopeptide producing organisms (24–28). However, it is mainly the genes of the *vanA*-type that are considered to have been mobilized from the original context and of these mainly *vanHAX* and *vanHBX* that have transferred widely either on transposons or conjugative plasmids. In this study, we also observed that the most commonly observed glycopeptide resistance genes were likely to be intrinsic chromosomal genes. Interestingly those intrinsic genes were not differentially abundant in Lynetten compared to the two other sewage systems, suggesting that their general environmental presence is not due to the selection by glycopeptide that may have been released from the pharmaceutical plants near Lynetten. The two most commonly observed forms of acquired glycopeptide resistance namely *vanHAX* and *vanHBX* were the only genes differentially abundant in Lynetten compared to the two other sewage treatment plants.

We were not able to confidently assign either *vanHAX* or *vanHBX* genes using Hi-C to any of the Hi-C clusters identified in our data. However, by mapping, the *vanHAX*- and *vanHBX*-containing reads to the conventional MAGs, we identified two species, both *S. parasuis*. Using BLASTN, we were able to assign *vanHAX* to four different species, while *vanHBX* could be assigned only to the phylum *Bacillota*.

While Hi-C sequencing is a powerful emerging technique for linking genes to their host organisms in metagenomes, it has limitations when it targets gene origins, including ARGs. One of the key challenges with Hi-C is contamination (50), where gene

clusters may appear to be associated with a single organism but are actually the result of chimeric associations formed during the proximity ligation process or simply due to the used bioinformatics methods. This contamination can influence the accurate assignment of resistance genes like *vanHAX* to specific bacterial species, especially in complex environmental samples like sewage where multiple species coexist. This was apparent here as most of the *vanHAX* and *vanHBX* reads could only confidently be assigned at higher taxonomic levels, and a number of the assignments to MAGs may have been influenced by contamination. These limitations suggest that future studies utilizing Hi-C in environmental AMR contexts should consider generating more extensive data and potentially integrating complementary methods, such as deeper sequencing or enhanced bioinformatic strategies, to mitigate these issues and improve the accuracy of taxonomic assignments.

The *vanHAX* gene cluster harbored on transposon *Tn1546* is one of the most commonly observed horizontally transmitted glycopeptide resistance genes. This gene cluster has been found in a number of different bacteria, mainly enterococcal species, but predominantly in *E. faecium* (21, 51–56). In a previous study using culturing only *vanA*-positive *E. faecium* was observed in Lynetten (20), even though different strains were observed and other species were detected from other environmental samples. In our study, we found *vanHAX* to be encoded on two different contigs, but were also only able to link the gene cluster to genomic contigs consistent with *vanHAX* being present in *E. faecium* and *E. saigonensis* in our conventional metagenomics approach, and in *E. faecium*, *E. faecalis*, *P. apiarius*, and *P. thiaminolyticus* using Hi-C. We found that half of the *vanHAX* Hi-C assembled contigs were identified as plasmids, while most *vanHBX* contigs were located on plasmids. These plasmids have been reported in enterococci species, consistent with the identified taxa. This, together with the small number of linked species, suggests that only limited horizontal gene transfer has taken place despite having colonized the sewer for more than 20 years. It cannot be excluded that a minor proportion of the sewer bacteria have acquired *vanHAX*, but then not to a degree where it was detectable using our approach. More sequencing would perhaps enable us to do so, but this would also come with an additional cost. It is well documented that *Paenibacillus* intrinsically harbors genes with homology to *vanHAX* (27). However, remapping the reads confirmed the presence of *vanHAX* in *Paenibacillus*. The reason for this update of *vanHAX* into presumed intrinsic glycopeptide-resistant bacterial species is unknown.

*vanHBX* has been observed in different enterococcal species, but most commonly in *E. faecalis*. We were not able to assign any of the *vanHBX* genes to a specific species, but only to the phylum Bacillota, which is still consistent with a presence in *Enterococcus*.

Sewage can serve as a reservoir for antimicrobial resistance. The presence of intrinsic glycopeptide resistance in the Lynetten sewage system could become a concern to human health if spillovers into natural waters occur. This could lead to dissemination of the established *vanHAX* and *vanHBX* gene clusters in natural environments.

We cannot exclude that our observations can be influenced by a continuous release of (glycopeptide-resistant) bacteria and/or glycopeptide or other antibiotics from the local producer of vancomycin. A detailed analysis for active antibiotics was not conducted in this study; however, future research should consider incorporating such analyses to provide a more comprehensive understanding of these influences.

This is, to the best of our knowledge, the first study evaluating the usefulness of Hi-C to study the distribution of glycopeptide resistance in sewage over time. While we were able to assign *vanHAX* and *vanHBX* reads to what seems like relevant bacterial phyla, families, and species, we also experienced that it in many cases was difficult to assign the non-*vanHAX* read to a specific bacterial species and that assignment to MAGs could be due to contamination of those MAGs. In many cases, we could only assign genera or family since the reads were for regions with limited taxonomic information. Thus, for future studies utilizing Hi-C it would likely be necessary to generate considerably more data, especially when investigating ARGs only found in low abundances.

## ACKNOWLEDGMENTS

This study was funded in part by the Novo Nordisk Foundation (Grant: NNF16OC0021856: Global Surveillance of Antimicrobial Resistance).

## AUTHOR AFFILIATIONS

[1]National Food Institute, Technical University of Denmark, Kgs Lyngby, Ghana
[2]Phase Genomics, Seattle, Washington, USA

## AUTHOR ORCIDs

Emilie Egholm Bruun Jensen ⓘ http://orcid.org/0000-0002-3214-7918
Saria Otani ⓘ http://orcid.org/0000-0002-2538-8086
Benjamin Auch ⓘ http://orcid.org/0000-0003-3125-5797
Frank M. Aarestrup ⓘ http://orcid.org/0000-0002-7116-2723

## FUNDING

| Funder | Grant(s) | Author(s) |
| --- | --- | --- |
| Novo Nordisk Fonden (NNF) | NNF16OC0021856 | Benjamin Auch |

## AUTHOR CONTRIBUTIONS

Emilie Egholm Bruun Jensen, Data curation, Formal analysis, Investigation, Methodology, Visualization, Writing – original draft | Saria Otani, Formal analysis, Investigation, Methodology, Resources, Supervision, Writing – review and editing | Ivan Liachko, Formal analysis, Investigation, Resources, Writing – review and editing | Benjamin Auch, Formal analysis, Resources, Writing – review and editing | Frank M. Aarestrup, Conceptualization, Data curation, Funding acquisition, Investigation, Project administration, Resources, Supervision, Writing – original draft

## DATA AVAILABILITY

The raw reads have been uploaded and are available at the European Nucleotide Archive (ENA). Samples collected between November 2015 and November 2018 were uploaded under PRJEB34633 and PRJEB13832 as part of a previous study investigating three different sewage treatment plants in Copenhagen: Rensningsanlæg Avedøre (RA), Rensningsanlæg Damhusåen (RD), and Rensningsanlæg Lynetten (RL) (2). Samples collected from Rensningsanlæg Lynetten (RL) from January 2019 to January 2023 were uploaded under PRJEB68319 and PRJEB79372 as part of a combined data set from five European cities, including metagenomics, qPCR, and Hi-C.

## ADDITIONAL FILES

The following material is available online.

### Supplemental Material

**Supplemental material (Spectrum01489-24-s0001.docx).** Figure S1; Tables S1 to S15.

### Open Peer Review

**PEER REVIEW HISTORY (review-history.pdf).** An accounting of the reviewer comments and feedback.

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
