## [Reviewer comments · Microbiology Spectrum]

Microbiology Spectrum

Using genomics to explore the epidemiology of vancomycin resistance in a sewage system

Emilie Jensen, Saria Otani, Ivan Liachko, Ben Auch, and Frank Aarestrup

Corresponding Author(s): Frank Aarestrup, Danmarks Tekniske Universitet

Review Timeline:

Submission Date:	June 19, 2024
Editorial Decision:	August 12, 2024
Revision Received:	September 18, 2024
Editorial Decision:	October 18, 2024
Revision Received:	October 22, 2024
Accepted:	November 12, 2024

Editor: Jinxin Liu

Reviewer(s): Disclosure of reviewer identity is with reference to reviewer comments included in decision letter(s). The following individuals involved in review of your submission have agreed to reveal their identity: Benjamin Cole Davis (Reviewer #2)

Transaction Report:

DOI: <https://doi.org/10.1128/spectrum.01489-24>

Re: Spectrum01489-24 (Using genomics to explore the epidemiology of vancomycin resistance in a sewage system)

Dear Prof. Frank M. Aarestrup:

Thank you for the privilege of reviewing your work. Below you will find my comments, instructions from the Spectrum editorial office, and the reviewer comments.

Revision Guidelines

Sincerely,
Jinxin Liu
Editor
Microbiology Spectrum

Reviewer #1 (Comments for the Author):

The authors investigated the possibility of spread of resistance (vanHAX) in an environment with historically high levels of antibiotic resistance, through a metagenomics approach. The study is well-conducted, timely, and of high relevance to better understand what drives selection and spread of antibiotic resistance.

A few minor additions would supplement an otherwise solid study.

- A brief discussion on how these genes commonly are mobilized and how spread is generated (e.g. what is favored in these species, conjugation, transformation, transduction), and has such spread between these species been documented before? You touch upon it in the discussion, but it would benefit to dive into this even further.

- Selection vs spread: it would be of interest to investigate the concentration of active antibiotics, or other stressors, in the wastewater at the different sites, and thus better understand the level of selection that the strains are exposed to. I do however understand that this falls outside of the scope for this specific project, but would appreciate a brief discussion in that regard.

Reviewer #2 (Comments for the Author):

The manuscript by Jensen et al. demonstrates an emerging metagenomic technique, Hi-C sequencing, to explore the genetic context of glycopeptide resistance in Danish sewage. This technique is valuable and provides much-needed contextual information for environmental AMR. However the authors need to further elaborate on the significance of the observations made, the methodology used, and improve on overall presentation/organization. Suggestions are made on how to do so in each subsection.

Intro 1st paragraph: Expand on what has been observed/learned in wastewater surveillance of AMR thus far. Why are you doing so in the first place? What are you trying to achieve? Use the general categories of "culture", "PCR-based", and "genomic" (or something very similar). Rephrase, ARGs are not "quantified" with culture techniques. Conventional shotgun metagenomics is semi-quantitative. Metagenomics allows for broad contextualization of genes with limitations, qPCR allows sensitivity. Use "limitation" instead of "drawback" with 10.7717/peerj.16695 as reference. Give explanation of what Hi-C sequencing is, how it's been used, and why it's advantageous to conventional techniques. What questions does it allow you to answer?

Intro 2nd paragraph: Van-R is ubiquitous in the environment and intrinsic to several Bacilli with wide ranges of relevance to human/animal health. Why is vanHAX/HBX important in relation to well-known vanA/B (and its dozens of homologs). What is ecological function, if any, of the van operon? Why might they be comfortable/selected/maintained in sewage (besides pharmaceutical pollution)? You want to study them because they were temporally stable, but the reader isn't communicated the significance.

Intro 3rd paragraph: Elaborate entirely on the van operon's structure as you're doing in-depth gene synteny analysis later. What does the reader need to know beforehand to be able to interpret the observations made on vanHAX/HBX flanking regions? I suggest having the conversation about vanHAX/HBX in the context of typical vanA/B gene synteny, including additional contextual analysis of their host range in your samples for comparison. Many van genes exist on pathogenicity islands and the most pernicious are on plasmids, for example. End the intro with stated hypotheses, objectives, and experimental approach. Samples and data: Giving the citation is not sufficient for explanation of sample collection and processing. Briefly recap how all samples were collected, processed, extracted, library prepped, and sequenced. Reference Table S1 and give the BioProject number for each sample.

Line 102: Give NovaSeq model, target insert size, read length, and targeted depth per sample.

Shotgun library prep, seq, and assembly: Blend this paragraph with the previous. Keep all library prep and sequencing information separate from the bioinformatic analysis. What do you mean samples were "normalized" with fastp?

Preprocessing and mapping of sequencing reads: What parameters were used with KMA? Explain that ResFinder and ResFinderFG28 are contained within PanRes. It wouldn't hurt to mention these are your guys' databases either.

Compositional data analysis: Define all terms. Screenshots of textbook excerpts are not sufficient. If you are going to use the equations, rewrite them in terms that are immediately relevant and interpretable to the dataset you are analyzing or else remove them. Consider turning this into a "stats" section adding all additional tests run, ggplot packages used, figure generation, etc.

Gene synteny and flanking regions: You haven't mentioned the importance of vanHAX/HBX in the intro. Start paragraph saying you're using the contigs for the analysis. Define "relevant thresholds".

Linking contigs to conventional metagenomic assembled genomes (MAGs): Give the binner used and cite the binner author, not a submitted manuscript. You don't need to give the KMA version or say PanRes database again. Make sure your completeness and contamination criteria meet standards (<https://www.nature.com/articles/nbt.3893>). Say you used the GTDB toolkit, not just the database. Say which reference tree they were placed in.

Linking Hi-C reads to Hi-C metagenomic clusters: Were the Hi-C reads quality checked at all? How were they assembled? If the clusters represent "individual organisms" why would they require decontamination in the first place? Which MAGpurify modules are being used to decontaminate the clusters and why? Specify "contigs" from "Hi-C contigs" throughout.

Linking Hi-C reads to conventional metagenomic assembled genomes (MAGs):

Clarify the meaning of "individual organisms" and elaborate on the workflows. You're mapping Hi-C reads to conventional contigs, pulling out just the van contigs, of which may or may not have been binned in that sample? In my mind, those bins/clusters are not "individual organisms", they are amalgams at the genera+ level or (if you're lucky) very closely related but not identical cells/haplotypes. Do only some of the contigs in said "clusters" have Hi-C linkages?

Linking vancomycin resistance to NCBI taxa: I highly suggest using mmseqs2's LCA algorithm with the GTDB for taxonomic annotation of contigs, it might resolve some of your ambiguities in Tables S6+7: <https://doi.org/10.1093/bioinformatics/btab184>. When addressing "reads" or "contigs" throughout make sure it's unambiguous whether they are normal or Hi-C, it's not clear in this paragraph.

Glycopeptide resistance profiles using the conventional metagenomics: Consider combining Table 1 and 2 and give more than 5 genes. I'd personally make Figure S1 into Figure 1 and put the Tables in the SI. If you say "more similar" compositionally, need to pair it with an ANOSIM (or similar tests), report the p-values.

Discussion: I wouldn't rule out the possibility of glycopeptide resistance evolution, mobilization, and selection in the receiving environment given the pharmaceutical pollution. They are more than likely chromosomal, but no plasmid prediction was attempted to rule it out. There needs to be a discussion on the significance of the vanHAX/HBX gene clusters found in sewage as it relates to human health, right now it's simply an observation of intrinsic glycopeptide resistance found in the environment. Putative explanations for the gene cluster temporal stability needs to be given with relation to other van genes and their hosts. Are these genes providing some advantage? There were also no real observations made about cross-treatment plant comparisons. I would also devote a heck of a lot more space discussing the benefits and drawbacks to Hi-C sequencing as this is an emerging technique in environmental AMR. Why are Hi-C gene clusters so contaminated if they're coming from a "individual organism"? What limitations does Hi-C have in this context? Lastly, I highly suggest performing some sort of plasmid analysis given that Hi-C sequencing is designed to do so (even if no van genes are found on said plasmids).

Figures: Make sure all text in-figure are larger and bolded. 300 dpi everything. Figures 4 and 5 are creative but remove the gray background box.

The manuscript by Jensen et al. demonstrates an emerging metagenomic technique, Hi-C sequencing, to explore the genetic context of glycopeptide resistance in Danish sewage. This technique is valuable and provides much-needed contextual information for environmental AMR. However the authors need to further elaborate on the significance of the observations made, the methodology used, and improve on overall presentation/organization. Suggestions are made on how to do so in each subsection.

Intro 1st paragraph: Expand on what has been observed/learned in wastewater surveillance of AMR thus far. Why are you doing so in the first place? What are you trying to achieve? Use the general categories of “culture”, “PCR-based”, and “genomic” (or something very similar). Rephrase, ARGs are not “quantified” with culture techniques. Conventional shotgun metagenomics is semi-quantitative. Metagenomics allows for broad contextualization of genes with limitations, qPCR allows sensitivity. Use “limitation” instead of “drawback” with 10.7717/peerj.16695 as reference. Give explanation of what Hi-C sequencing is, how it’s been used, and why it’s advantageous to conventional techniques. What questions does it allow you to answer?

Intro 2nd paragraph: Van-R is ubiquitous in the environment and intrinsic to several Bacilli with wide ranges of relevance to human/animal health. Why is vanHAX/HBX important in relation to well-known vanA/B (and its dozens of homologs). What is ecological function, if any, of the van operon? Why might they be comfortable/selected/maintained in sewage (besides pharmaceutical pollution)? You want to study them because they were temporally stable, but the reader isn’t communicated the significance.

Intro 3rd paragraph: Elaborate entirely on the van operon’s structure as you’re doing in-depth gene synteny analysis later. What does the reader need to know beforehand to be able to interpret the observations made on vanHAX/HBX flanking regions? I suggest having the conversation about vanHAX/HBX in the context of typical vanA/B gene synteny, including additional contextual analysis of their host range in your samples for comparison. Many van genes exist on pathogenicity islands and the most pernicious are on plasmids, for example. End the intro with stated hypotheses, objectives, and experimental approach.

Samples and data: Giving the citation is not sufficient for explanation of sample collection and processing. Briefly recap how all samples were collected, processed, extracted, library prepped, and sequenced. Reference Table S1 and give the BioProject number for each sample.

Line 102: Give NovaSeq model, target insert size, read length, and targeted depth per sample.

Shotgun library prep, seq, and assembly: Blend this paragraph with the previous. Keep all library prep and sequencing information separate from the bioinformatic analysis. What do you mean samples were “normalized” with fastp?

Preprocessing and mapping of sequencing reads: What parameters were used with KMA? Explain that ResFinder and ResFinderFG28 are contained within PanRes. It wouldn’t hurt to mention these are your guys’ databases either.

Compositional data analysis: Define all terms. Screenshots of textbook excerpts are not sufficient. If you are going to use the equations, rewrite them in terms that are immediately relevant and interpretable to the dataset you are analyzing or else remove them. Consider turning this into a “stats” section adding all additional tests run, ggplot packages used, figure generation, etc.

Gene synteny and flanking regions: You haven’t mentioned the importance of vanHAX/HBX in the intro. Start paragraph saying you’re using the contigs for the analysis. Define “relevant thresholds”.

Linking contigs to conventional metagenomic assembled genomes (MAGs): Give the binner used and cite the binner author, not a submitted manuscript. You don’t need to give the KMA version or say PanRes database again. Make sure your completeness and contamination criteria meet standards (<https://www.nature.com/articles/nbt.3893>). Say you used the GTDB toolkit, not just the database. Say which reference tree they were placed in.

Linking Hi-C reads to Hi-C metagenomic clusters: Were the Hi-C reads quality checked at all? How were they assembled? If the clusters represent “individual organisms” why would they require decontamination in the first place? Which MAGpurify modules are being used to decontaminate the clusters and why? Specify “contigs” from “Hi-C contigs” throughout.

Linking Hi-C reads to conventional metagenomic assembled genomes (MAGs):

Clarify the meaning of “individual organisms” and elaborate on the workflows. You’re mapping Hi-C reads to conventional contigs, pulling out just the *van* contigs, of which may or may not have been binned in that sample? In my mind, those bins/clusters are not “individual organisms”, they are amalgams at the genera+ level or (if you’re lucky) very closely related but not identical cells/haplotypes. Do only some of the contigs in said “clusters” have Hi-C linkages?

Linking vancomycin resistance to NCBI taxa: I highly suggest using mmseqs2’s LCA algorithm with the GTDB for taxonomic annotation of contigs, it might resolve some of your ambiguities in Tables S6+7: <https://doi.org/10.1093/bioinformatics/btab184>. When addressing “reads” or “contigs” throughout make sure it’s unambiguous whether they are normal or Hi-C, it’s not clear in this paragraph.

Glycopeptide resistance profiles using the conventional metagenomics: Consider combining Table 1 and 2 and give more than 5 genes. I’d personally make Figure S1 into Figure 1 and put the Tables in the SI. If you say “more similar” compositionally, need to pair it with an ANOSIM (or similar tests), report the p-values.

Discussion: I wouldn’t rule out the possibility of glycopeptide resistance evolution, mobilization, and selection in the receiving environment given the pharmaceutical pollution. They are more than likely chromosomal, but no plasmid prediction was attempted to rule it out. There needs to be a discussion on the significance of the vanHAX/HBX gene clusters found in sewage as it relates to human health, right now it’s simply an observation of intrinsic glycopeptide resistance found in the environment. Putative explanations for the gene cluster temporal stability needs to be given with relation to other van genes and their hosts. Are these genes providing some advantage? There were also no real observations made about cross-treatment plant comparisons. I would also devote a heck of a lot more

space discussing the benefits and drawbacks to Hi-C sequencing as this is an emerging technique in environmental AMR. Why are Hi-C gene clusters so contaminated if they're coming from a "individual organism"? What limitations does Hi-C have in this context? Lastly, I highly suggest performing some sort of plasmid analysis given that Hi-C sequencing is designed to do so (even if no van genes are found on said plasmids).

Figures: Make sure all text in-figure are larger and bolded. 300 dpi everything. Figures 4 and 5 are creative but remove the gray background box.

Reviewer #1 (Comments for the Author):

The authors investigated the possibility of spread of resistance (vanHAX) in an environment with historically high levels of antibiotic resistance, through a metagenomics approach. The study is well-conducted, timely, and of high relevance to better understand what drives selection and spread of antibiotic resistance.

A few minor additions would supplement an otherwise solid study.

- A brief discussion on how these genes commonly are mobilized and how spread is generated (e.g. what is favored in these species, conjugation, transformation, transduction), and has such spread between these species been documented before? You touch upon it in the discussion, but it would benefit to dive into this even further.

“We have briefly included a sentence about this in the discussion (lines 374-378). We know the Tn1546 and Tn1549 transposons can transfer, and as far as we know all the plasmids they can be located on so far are investigated. However, Enterococci often generate fusion-plasmids with several replicons making assignment difficult. Furthermore, what has mainly been observed in Denmark on the clinical side is clonal replacement (with new plasmids) rather than horizontal transfer. This is however, outside the scope of this study.”

- Selection vs spread: it would be of interest to investigate the concentration of active antibiotics, or other stressors, in the wastewater at the different sites, and thus better understand the level of selection that the strains are exposed to. I do however understand that this falls outside of the scope for this specific project, but would appreciate a brief discussion in that regard.

“This is a good point. We have included a sentence in the discussion about this.”

Reviewer #2 (Comments for the Author):

The manuscript by Jensen et al. demonstrates an emerging metagenomic technique, Hi-C sequencing, to explore the genetic context of glycopeptide resistance in Danish sewage. This technique is valuable and provides much-needed contextual information for environmental AMR. However the authors need to further elaborate on the significance of the observations made, the methodology used, and improve on overall presentation/organization. Suggestions are made on how to do so in each subsection.

Intro 1 st paragraph: Expand on what has been observed/learned in wastewater surveillance of AMR thus far. Why are you doing so in the first place? What are you trying to achieve? Use the general categories of “culture”, “PCR-based”, and “genomic” (or something very similar). Rephrase, ARGs are not “quantified” with culture techniques. Conventional shotgun metagenomics is semi-quantitative. Metagenomics allows for broad contextualization of genes with limitations, qPCR allows sensitivity.

“We have changed this text accordingly.”

Use "limitation" instead of "drawback" with 10.7717/peerj.16695 as reference.

"We have done so and included the reference (line 56)."

Give explanation of what Hi-C sequencing is, how it's been used, and why it's advantageous to conventional techniques. What questions does it allow you to answer?

"We have added text in the introduction on this (lines 59-63)."

Intro 2nd paragraph: Van-R is ubiquitous in the environment and intrinsic to several Bacilli with wide ranges of relevance to human/animal health. Why is vanHAX/HBX important in relation to well-known vanA/B (and its dozens of homologs).

"We have included a brief paragraph about the clinical surveillance in Denmark (line 74-76)."

What is ecological function, if any, of the van operon?

"We have absolutely no idea."

Why might they be comfortable/selected/maintained in sewage (besides pharmaceutical pollution)?

"We do not know and it was also outside our scope. We do, however, mention this in the discussion."

You want to study them because they were temporally stable, but the reader isn't communicated the significance.

"Correct, we have adjusted the text to explain this better in the introduction (lines 89-93)."

Intro 3rd paragraph: Elaborate entirely on the van operon's structure as you're doing in-depth gene synteny analysis later. What does the reader need to know beforehand to be able to interpret the observations made on vanHAX/HBX flanking regions? I suggest having the conversation about vanHAX/HBX in the context of typical vanA/B gene synteny, including additional contextual analysis of their host range in your samples for comparison. Many van genes exist on pathogenicity islands and the most pernicious are on plasmids, for example.

"We have included more information on this in the introduction (lines 78-84). However, we would also like to avoid making this too much of a review paper while providing the needed information only for the scope of this research."

End the intro with stated hypotheses, objectives, and experimental approach.

"We have included a sentence stating our hypothesis (lines 91-93)."

Samples and data: Giving the citation is not sufficient for explanation of sample collection and processing. Briefly recap how all samples were collected, processed, extracted, library prepped, and sequenced. Reference Table S1 and give the BioProject number for each sample.

“A brief description on how the samples were prepared, handled and sequenced was added to the main text (lines 108-111, 129-131). We have added the ENA Run Accession for each sample in Supplementary table 1.”

Line 102: Give NovaSeq model, target insert size, read length, and targeted depth per sample.

“Details on the NovaSeq platform and its output were added (lines 129-131).”

Shotgun library prep, seq, and assembly: Blend this paragraph with the previous. Keep all library prep and sequencing information separate from the bioinformatic analysis. What do you mean samples were “normalized” with fastp?

“We merged all the wet lab prep. Together and separated it from the bioinformatics analyses. Normalized with fastp has been replaced with trimmed using fastp which is the more accurate description (lines 142).”

Preprocessing and mapping of sequencing reads: What parameters were used with KMA? Explain that ResFinder and ResFinderFG28 are contained within PanRes. It wouldn't hurt to mention these are your guys' databases either.

“The parameters used for KMA have been added (line 145). Additionally, we have added a sentence explaining that ResFinder and ResFinderFG are part of the PanRes database.”

Compositional data analysis: Define all terms. Screenshots of textbook excerpts are not sufficient. If you are going to use the equations, rewrite them in terms that are immediately relevant and interpretable to the dataset you are analyzing or else remove them. Consider turning this into a “stats” section adding all additional tests run, ggplot packages used, figure generation, etc.

“We have elaborated on the centred-log ratio (clr) transformation with an explanation related directly to our analysis (lines 162-164).”

Gene synteny and flanking regions: You haven't mentioned the importance of vanHAX/HBX in the intro. Start paragraph saying you're using the contigs for the analysis. Define “relevant thresholds”.

“The manuscript has been adjusted accordingly. The paragraph now starts with a statement about the analysis being based on the contigs (line 172). The relevant threshold has been elaborated by adding a reference to Supplementary table 2, which contains the thresholds used (lines 177-178).”

Linking contigs to conventional metagenomic assembled genomes (MAGs): Give the binner used and cite the binner author, not a submitted manuscript. You don't need to give the KMA version or say PanRes database again.

"We have added information on the trimming and adapter removal, assembly process and binning tool used (line 184)."

Make sure your completeness and contamination criteria meet standards (<https://www.nature.com/articles/nbt.3893>).

"We share the reviewer's opinion on the importance of following the Minimum Information about a Metagenome-Assembled Genome (MIMAG) from the Genomics Standard Consortium. We followed the MIMAG for the definition of the High Quality and Medium Quality bins, with the latter having a more strict threshold than proposed in MIMAG (completeness ≥ 70 instead of completeness ≥ 50)."

Say you used the GTDB toolkit, not just the database. Say which reference tree they were placed in.

"GTDB-Tk was added (line 191), as well as a further explanation on the reference tree."

Linking Hi-C reads to Hi-C metagenomic clusters: Were the Hi-C reads quality checked at all? How were they assembled?

"We have added a paragraph about the handling and analysis of the Hi-C reads (142-143)."

If the clusters represent "individual organisms" why would they require decontamination in the first place?

"We have rephrased this (line 198). It is a known issue that some bins contain contamination. This is what is addressed by running CheckM. Since we observed a high contamination in certain bins, these were attempted to be decontaminated."

Which MAGpurify modules are being used to decontaminate the clusters and why?

"We have added some text about the flags used for the MAGpurify analysis (lines 200-204)."

Specify "contigs" from "Hi-C contigs" throughout.

We have specified Hi-C contigs throughout the paragraph."

Linking Hi-C reads to conventional metagenomic assembled genomes (MAGs):

Clarify the meaning of "individual organisms" and elaborate on the workflows. You're mapping Hi-C reads to conventional contigs, pulling out just the van contigs, of which may or may not have been binned in that sample? In my mind, those bins/clusters are not "individual organisms", they are amalgams at the genera+ level or (if you're lucky) very closely related but not identical cells/haplotypes. Do only some of the contigs in said "clusters" have Hi-C linkages?

“We agree with the observation made by the reviewer, and we have rephrased the sentence about Hi-C metagenomic clusters (line 195). We have elaborated on the workflow (lines 204-205) as well as the idea behind the analysis (lines 207-208). To not limit the taxonomic analysis to Hi-C reads being binned in Hi-C clusters, we included all Hi-C reads containing vancomycin resistance and mapped them to the conventional MAGs.”

Linking vancomycin resistance to NCBI taxa: I highly suggest using mmseqs2’s LCA algorithm with the GTDB for taxonomic annotation of contigs, it might resolve some of your ambiguities in Tables S6+7: <https://doi.org/10.1093/bioinformatics/btab184>.

“We thank the reviewer for this suggestion. We applied the mmseqs2 taxonomy workflow using the LCA algorithm with the GTDB database for taxonomic annotation of the vancomycin-containing contigs (--lca mode 4). Unfortunately this did not resolve the ambiguities observed for these contigs to a high degree. However, we were able to resolve the taxonomic classification of 2 *vanHAX*-containing contigs, and we have added this to the results (lines 227-230, figure 4, figure 5, supplementary table 5, supplementary table 5, and supplementary table 7).”

When addressing “reads” or “contigs” throughout make sure it’s unambiguous whether they are normal or Hi-C, it’s not clear in this paragraph.

“We have addressed this throughout the manuscript, using the term “sequences” when talking about both Hi-C reads and conventional contigs.”

Glycopeptide resistance profiles using the conventional metagenomics: Consider combining Table 1 and 2 and give more than 5 genes. I’d personally make Figure S1 into Figure 1 and put the Tables in the SI.

“Table 1 includes the top 5 most abundant *van* resistance genes from the ResFinder db. Expanding this will only include non-abundant resistance genes (clr median), which does not add to the analysis of the most abundant resistance genes. We have expanded Table 2 to include top 10, since these still have a high abundance (high clr median) across the samples.”

If you say “more similar” compositionally, need to pair it with an ANOSIM (or similar tests), report the p-values.

“We have added a PERMANOVA test to support the statement (line 269-270).”

Discussion: I wouldn’t rule out the possibility of glycopeptide resistance evolution, mobilization, and selection in the receiving environment given the pharmaceutical pollution. They are more than likely chromosomal, but no plasmid prediction was attempted to rule it out.

“We thank the reviewer for suggesting the additional analysis, as we agree this will add to the overall analysis and understanding of glycopeptide resistance in the sewer system. We have added a plasmid analysis on both the Hi-C assembled plasmids, the Hi-C assembled contigs and the conventional contigs (lines 240-246, lines 348-363). We have added a section to the discussion as well (lines 416-421)”

There needs to be a discussion on the significance of the vanHAX/HBX gene clusters found in sewage as it relates to human health, right now it's simply an observation of intrinsic glycopeptide resistance found in the environment.

“We have added a paragraph discussing the significance of the gene clusters found and how it related to human health (lines 433-436)”

Putative explanations for the gene cluster temporal stability needs to be given with relation to other van genes and their hosts. Are these genes providing some advantage?

“We cannot exclude that a continuous release of antibiotics from the producer might select for resistance. However, analysis of active antibiotics in the sewage was not conducted in this study. We have added this as a consideration for future research (lines 438-442).”

There were also no real observations made about cross-treatment plant comparisons.

“We have added this (line 384).”

I would also devote a heck of a lot morespace discussing the benefits and drawbacks to Hi-C sequencing as this is an emerging technique in environmental AMR. Why are Hi-C gene clusters so contaminated if they're coming from a “individual organism”? What limitations does Hi-C have in this context?

“We have added a paragraph on Hi-C output and its efficacy in the discussion now (lines 393-405)”

Lastly, I highly suggest performing some sort of plasmid analysis given that Hi-C sequencing is designed to do so (even if no van genes are found on said plasmids).

“We thank the reviewer for this suggestion. We performed a plasmid analysis and found instances of *vanHAX* and *vanHBX* resistance genes in plasmids reported in Enterococci species. We have updated the manuscript with the new results throughout the result section.”

Figures: Make sure all text in-figure are larger and bolded. 300 dpi everything. Figures 4 and 5 are creative but remove the gray background box.

“We have removed the grey box from figure 4 and 5. All figures will be uploaded in high quality for the submitted manuscript.”

Re: Spectrum01489-24R1 (Using genomics to explore the epidemiology of vancomycin resistance in a sewage system)

Dear Prof. Frank M. Aarestrup:

Thank you for the privilege of reviewing your work. Below you will find my comments, instructions from the Spectrum editorial office, and the reviewer comments.

Revision Guidelines

Sincerely,
Jinxin Liu
Editor
Microbiology Spectrum

Reviewer #1 (Comments for the Author):

The authors have addressed all the raised concerns.

Reviewer #2 (Comments for the Author):

The authors have done a wonderful job revising the manuscript, the presentation and clarity of the science has greatly improved. I caught only a few small mistakes/made a few suggestions below. All lines refer to the "Spectrum01489-24R1-Merged_PDF" document.

Line 62: *can infer the...

Line 89-90: ...we wanted to investigate | we investigated...

Line 92: ...the Lynetten sewer, *substantial evolution...

Lines 163-165: remove "were examined"... using *the Flankophile pipeline.

Line 193: *phylo-markers

Lines 231-234: Rephrase paragraph to explain how Hi-C assembled plasmids were first identified within the set of contigs/bins and then annotated with KMA looking for van genes.

Line 360-361: Clarify the end of the sentence. Are you saying that the glycopeptide resistance existed prior to the factory or that it originated from the factory itself?

Line 366: either as transposons | either *on transposons

Line 373: Differentially*

Line 383: There's a "47" in text that needs to be removed.

Lines 431-434: Missing a comma after "...families and species"

For Figure 2, the gene labels and legend text need to be larger. The rest of the figures look great.

Reviewer #1 (Comments for the Author):

The authors have addressed all the raised concerns.

Reviewer #2 (Comments for the Author):

The authors have done a wonderful job revising the manuscript, the presentation and clarity of the science has greatly improved. I caught only a few small mistakes/made a few suggestions below. All lines refer to the "Spectrum01489-24R1-Merged_PDF" document.

Line 62: *can infer the...

"We have added "can" to the sentence (line 61)."

Line 89-90: ...we wanted to investigate | we investigated...

"We have rephrased this (lines 86-87)."

Line 92: ...the Lynetten sewer, *substantial evolution...

"We have corrected this (line 89)."

Lines 163-165: remove "were examined"... using *the Flankophile pipeline.

"We have rephrased this (line 162)."

Line 193: *phylo-markers

"We have corrected this (line 190)."

Lines 231-234: Rephrase paragraph to explain how Hi-C assembled plasmids were first identified within the set of contigs/bins and then annotated with KMA looking for van genes.

"We have added this (lines 228-229)."

Line 360-361: Clarify the end of the sentence. Are you saying that the glycopeptide resistance existed prior to the factory or that it originated from the factory itself?

"We have rephrased this (lines 356-357)."

Line 366: either as transposons | either *on transposons

"We have rephrased this (line 362)."

Line 373: Differentially*

"We have added this (line 368)."

Line 383: There's a "47" in text that needs to be removed.

"This has been removed (line 379)."

Lines 431-434: Missing a comma after "...families and species"

"We have adjusted this (line 428)."

For Figure 2, the gene labels and legend text need to be larger. The rest of the figures look great.

"We have increased this (figure 2)."

Re: Spectrum01489-24R2 (Using genomics to explore the epidemiology of vancomycin resistance in a sewage system)

Dear Prof. Frank M. Aarestrup:

Your manuscript has been accepted, and I am forwarding it to the ASM production staff for publication. Your paper will first be checked to make sure all elements meet the technical requirements. ASM staff will contact you if anything needs to be revised before copyediting and production can begin. Otherwise, you will be notified when your proofs are ready to be viewed.

Sincerely,
Jinxin Liu
Editor
Microbiology Spectrum